# Knowledge, Attitudes, and Practices of Antibiotic Use and AMR in Low-Income Urban Delhi, India: A Community-Based Cross-Sectional Study

**DOI:** 10.3390/antibiotics14121184

**Published:** 2025-11-21

**Authors:** Shivani Rao, Saurav Basu, Srishty Rajaura, Mrunali Zode, Mongjam Meghachandra Singh, Nandini Sharma

**Affiliations:** 1Department of Community Medicine, Maulana Azad Medical College, New Delhi 110002, India; shivani.rao@delhi.gov.in (S.R.); srishtyrajaura2530@gmail.com (S.R.); 2Department of Community Medicine, ESI-PGIMSR, ESIC Medical College, Joka, Kolkata 700104, India; 3Indian Council of Medical Research, New Delhi 110029, India; mzode04@gmail.com; 4Department of Community Medicine, SGT University, Gurugram 122505, India; hod.community@sgtuniversity.org

**Keywords:** antimicrobial resistance (AMR), antibiotic stewardship, antibiotic awareness, health literacy, self-medication

## Abstract

Background: Antimicrobial resistance (AMR) is a major public health threat in India, driven by the misuse of antibiotics. However, there is a lack of community-level data on AMR awareness and antibiotic use practices, especially in low-income urban settings. This study aimed to assess the knowledge, attitudes, and practices (KAPs) regarding antibiotic use and AMR among residents of a low-income area in Delhi. Methods: A community-based, cross-sectional study was conducted from April to June 2025 among 1601 adults in a low-income urban area of Delhi, using a validated questionnaire. Data on sociodemographic and KAP regarding antibiotics and AMR were collected. Ordinal and binary logistic regression analyses were performed to identify sociodemographic determinants of KAP. Results: While 68.0% of participants had heard of antibiotics, specific knowledge was poor. Only 19.4% were aware of AMR, and significant misconceptions were prevalent, with 47.5% believing that antibiotics were effective for colds and flu. Self-medication with antibiotics was reported by 24.6% of respondents. Lower educational attainment, female gender, and lower income were significantly associated with poorer antibiotic knowledge and practices. Higher knowledge scores were strongly associated with better practices (AOR: 1.30, 95% CI: 1.23, 1.37, *p* < 0.001). Conclusions: Critical knowledge gaps and high-risk antibiotic use practices are prevalent in this low-income urban community, strongly linked to social determinants like education, gender, and income. There is an urgent need for evidence-informed and tailored educational and behavior change interventions to curb irrational antibiotic demand and preserve these vital medicines.

## 1. Introduction

Antimicrobial resistance (AMR) is the process by which microorganisms—including bacteria, viruses, fungi, and parasites—evolve mechanisms to withstand the effects of antimicrobial drugs, such as antibiotics, that were previously effective for treatment [1]. While AMR is a natural evolutionary process, its alarming acceleration is overwhelmingly driven by the intense selective pressure exerted by the misuse and overuse of antimicrobial agents, particularly antibiotics [2,3].

The detrimental impact of AMR affects individual health outcomes and public health systems as resistant infections prolong hospitalizations, increase medical costs, and have higher mortality rates. The lack of effective antibiotics also threatens the success of modern medicine, including surgeries and cancer therapies [4,5]. The World Health Organization (WHO) identifies AMR as one of the top ten global public health threats—driven by poor infection control, irrational prescription, and a lack of new drug development, with projections in 2050 estimating 1.91 million (1.56–2.26) deaths attributable to AMR and 8.22 million (6.85–9.65) deaths associated with AMR [1,2].

There has been growing recognition over the past decade of the complex interplay of public knowledge, attitudes, and practices (KAPs) regarding antibiotics and AMR. Widespread public misconceptions, such as the belief in antibiotic efficacy in viral infections, the discontinuation of prescribed courses upon symptomatic improvement, and a lack of awareness of antibiotic misuse driving antibiotic resistance, can be linked to these inappropriate antibiotic use practices [6,7,8].

A 2015 WHO multi-country survey revealed significant misconceptions regarding antimicrobial resistance (AMR), with 76% of respondents incorrectly believing that resistance develops in the human body rather than in bacteria. Nearly two in three respondents assumed that correct antibiotic usage (66%) or infrequent use (44%) mitigates their individual vulnerability to drug-resistant infections. A majority of respondents felt disempowered and felt that they could do little to stop AMR (57%) and instead expected solutions to emerge from the medical community (64%) [9].

India, the country with the highest antibiotic consumption globally, faces major challenges in combating this “silent pandemic” [10]. Improper antibiotic use across clinical, veterinary, and agricultural settings drives the selection and proliferation of resistant microorganisms [11]. Contributing factors to this problem are the over-prescription of antibiotics by medical professionals and disproportionate empirical usage, poor patient adherence to prescribed treatment regimens, and the widespread practice of antibiotic self-medication without a valid prescription [12,13,14].

Studies on the general population in India reveal very low levels of awareness of antibiotics and rampant antibiotic misuse. However, there is a significant paucity of data, especially in the post-COVID-19 scenario. A survey conducted in rural villages of the Eastern state of Odisha (2023) showed that only 44.7% of participants had heard of antibiotic medicines, while one in five discontinued antibiotics on symptomatic recovery [15]. Another study from the central Indian city of Ujjain using a convenient sample of people from a mass gathering observed that 97% of the participants had poor knowledge about antibiotics, and 88.5% exhibited inappropriate practices, such as sharing antibiotics and not completing prescribed courses [16]. A study from coastal South India found that nearly half of the respondents believed that antibiotics killed viruses and used them for common colds [17].

In the Indian context, there remains a significant lack of granular, community-level data on the prevalence of AMR awareness and antibiotic use practices within socioeconomically deprived populations such as residents of urban slums and resettlement colonies. The National Policy on AMR containment in India has recommended prioritizing antibiotic and AMR-related awareness in the general population, which requires evidence-based public health campaigns [18]. Therefore, it is crucial to understand specific sociodemographic factors such as income, education level, gender, and household characteristics that are associated with poor knowledge and inappropriate practices in this high-risk environment.

Therefore, we conducted this study with the objectives of determining the awareness of antimicrobial resistance (AMR) and antibiotic use practices and their sociodemographic determinants among residents of low-income areas in Delhi.

## 2. Results

### 2.1. Participant Characteristics

A total of 1601 participants aged 18 to 80 years were included in the study. The majority (61.9%) were young adults aged 18–35 years, while 124 (7.7%) belonged to the geriatric age-group (age ≥ 60 years). The sample was characterized by varied educational attainment, with a significant minority (32%) not having completed secondary education (XII grade) (Table 1).

### 2.2. Knowledge and Misconceptions Regarding Antibiotics and AMR

Significant gaps in knowledge regarding antibiotics and antimicrobial resistance (AMR) were identified. Although 68.0% of participants reported that they knew about antibiotics, specific knowledge was limited. Only 34.7% correctly stated that antibiotics must be taken for the full prescribed duration. Concerningly, 23.7% considered sharing antibiotics with friends or family for similar symptoms to be acceptable, and 24.2% indicated they would repurchase antibiotics or request them from a doctor for a similar illness, provided the treatment was perceived to be effective previously. Misconceptions about the indications for antibiotic use were prevalent. When presented with a list of medical conditions, only a minority of respondents correctly identified bladder/urinary tract infections (18.4%), skin/wound infections (22.5%), and gonorrhea (1.1%) as treatable with antibiotics. Conversely, substantial proportions incorrectly believed that antibiotics are effective for conditions such as colds and flu (47.5%), sore throats (20.0%), and headaches (10.6%).

Awareness of AMR was notably low. Only 19.4% (*n* = 310) recognized any related terms (e.g., ‘antibiotic resistance’, ‘superbugs’, ‘AMR’), while 80.6% had never encountered them. Moreover, a majority of respondents were uncertain about the mechanisms of resistance (58.8%), its transmissibility (54.9%), and its potential impact on other medical procedures (65.5%) (Table 2).

### 2.3. Antibiotic Use and Self-Medication Practices

Regarding antibiotic utilization, 47.7% of participants reported that their most recent course of antibiotics was obtained via a prescription from a healthcare provider, and 48.6% received instructions on proper usage. Self-medication with antibiotics was reported by 24.6% of respondents, most commonly for symptoms such as fever, pain, cough, and headache. The majority (60.9%) reported checking medication expiry dates (Table 3).

### 2.4. Sociodemographic Correlates of Knowledge, Attitudes, and Practices Related to Antibiotics

Generalized ordinal regression analysis revealed that females had significantly lower odds of having higher knowledge compared to males (AOR: 0.82; 95% CI: 0.67, 1.00). Educational attainment was a strong predictor of knowledge; participants with no high school degree had substantially lower odds of higher knowledge compared to those with a college degree or above (AOR: 0.16; 95% CI: 0.11, 0.22). A lower monthly income (<₹10,000) was also associated with lower odds of higher knowledge (AOR: 0.21; 95% CI: 0.12, 0.37). In contrast, participants with no high school degree had higher odds of reporting a more positive attitude (AOR: 2.36; 95% CI: 1.63, 3.41), while those with lower income had lower odds of a positive attitude (AOR: 0.10; 95% CI: 0.06, 0.17). Finally, higher knowledge was favorably associated with more positive attitudes (AOR: 1.28; 95% CI: 1.21, 1.35).

In the adjusted model, educational attainments and income were significantly associated with antibiotic use practices. Participants with lower educational attainments (below high school) (AOR: 0.57, 95% CI: 0.35, 0.93, *p* = 0.024) and high school only (AOR: 0.28, 95% CI: 0.14, 0.58, *p* = 0.001) had significantly lower odds of engaging in better practices compared to those with high educational attainments (college degree and above), although having 12th grade education showed no statistically significant difference. Households with lower monthly income (less than ₹10,000, ₹10,001–₹25,000, or ₹25,001–₹50,000) were likely to have lower odds of better practices compared to households with an income above ₹50,000. Importantly, higher knowledge scores were positively associated with better practices (AOR: 1.30, 95% CI: 1.23, 1.37, *p* < 0.001), whereas higher attitude scores were inversely associated with good practices (AOR: 0.93, 95% CI: 0.92, 0.95, *p* < 0.001) (Table 4).

## 3. Discussion

This community-based, cross-sectional study identifies major awareness deficits and high-risk antibiotic use patterns and practices among low-income communities in Delhi that are contributing to fueling the countrywide AMR crisis. The study findings reveal profound gaps in antibiotic knowledge, high rates of self-medication, and poor antibiotic medication adherence, which strongly correlate with low education and household income. This complex interplay of severe antibiotic misuse with adverse social determinants, particularly high infectious disease burden, low socioeconomic status, poor health literacy, and challenges in healthcare access, is indicative of an ongoing AMR syndemic [19].

In this study, less than one in five participants had heard about AMR, reflecting an alarmingly low awareness, which is in concordance with past studies from India and other low- and middle-income countries [8,15,16,17]. Globally, a systematic review and meta-analysis by Gualano et al. (2015) reported that 33.7% were unaware that antibiotics treat bacterial infections, while 26.9% were unaware that antibiotic misuse can lead to this problem [7]. Furthermore, a majority of participants in studies from Jordan, New Zealand, the midwestern USA, and South Korea were unaware that antibiotics treat bacterial infections, while majorities in South Korea, Jordan, Malaysia, and New York City’s Latino community incorrectly believed antibiotics treat the cold and flu [7]. In this study, while nearly two in three participants reported ‘knowing about antibiotics’, barely one in three were aware of the necessity of completing a course. Similarly, nearly half did not understand the ineffectiveness of antibiotics against viral illnesses, which echoes findings from the WHO 2015 Multi-country study [9]. The awareness levels of the participants in the present study are even lower compared to urban slum dwellers in Uganda, wherein good knowledge was observed in nearly half (47.7%) of the respondents [20].

This study affirms the persistence of a significant antibiotic knowledge deficit within a dense urban-slum and resettlement colony population. Notably, while 79.8% of respondents had completed at least a high-school education, their knowledge gaps remained. The educational attainment of our sample is considerably higher than India’s current adult literacy rate (80.9%, per the 2023-24 Periodic Labour Force Survey) [21]. We attribute this discrepancy to the lower proportion of geriatric participants in our study compared to the national average, as this demographic typically has less formal education and overall lower educational attainments.

The knowledge gaps observed in the participants may also be compounded by barriers to accessing conventional public health messaging and failure to tailor public health campaigns, resulting in a potential lack of effective outreach. Furthermore, in line with the findings of Muflih et al. (2021), this may also suggest that many individuals lack the functional health literacy required to obtain, process, and comprehend information regarding the rational use of antibiotics [22]. Consequently, there is a need to reexamine the effectiveness of conventional public health messaging in improving awareness of AMR in the general population, considering the low perceived susceptibility of this grave public health threat [10,23]. Moreover, the reinforcement of cognitive biases through coincidental recovery of individuals after taking antibiotics for mostly self-limiting conditions such as the common cold, flu, and diarrhea can promote a spurious belief in antibiotic efficacy and drive patients into exerting social pressure on physicians to prescribe antibiotics [24,25].

In this study, a majority of the participants reported high-risk behaviors likely contributing to antibiotic misuse. Over a third of the participants admitted to self-medicating with antibiotics, which is comparatively lower compared to the global pooled average (43%), comparable to European countries (34.7%), but higher than the prevalence in Asia (25.7%) [26]. However, the practice of antibiotic self-medication in this study is higher compared to other Indian studies in rural settings [15] and through online surveys [27]. This phenomenon can be attributed to the ease of accessing antibiotics without a formal prescription from local pharmacies in the study setting, with the majority of the participants sourcing antibiotics without consulting formal health practitioners.

In the present study, a strong, graded relationship between lower educational attainment and poorer antibiotic-related KAP was observed, suggesting its correlation with health literacy [22]. Further, female participants compared to male participants had significantly lower odds of higher antibiotic KAP, suggestive of gender inequities likely driving antibiotic misuse in vulnerable populations. Future studies should also evaluate the decision-making power of mothers and female relatives to understand in-depth gender roles in fueling antibiotic misuse, particularly among children. Although lower educational attainments were associated with a positive attitude despite poorer knowledge of AMR, it is probably reflective of an attitude of compliance with medical authorities. This study, therefore, adds that a positive attitude without adequate knowledge of antibiotics, especially in healthcare-deficient areas, can also inadvertently promote antibiotic misuse, such as through recommendations from unlicensed practitioners or the practice of self-medication for an individual or any other household member.

### Strengths and Limitations

The primary strength of this study lies in its community-based design and large sample size, and a random sample which was drawn from a dense urban agglomerate and socioeconomically vulnerable population at high risk of infections. The study findings are therefore likely to be generalizable to similar low-income settings in cities across Northern India. However, this study has certain limitations. First, its cross-sectional nature precludes any inference of causality between sociodemographic factors and antibiotic KAP. Second, the data on antibiotic use and practices were self-reported and may be subject to recall and social desirability biases. Participants may have under-reported perceived socially undesirable practices such as self-medication. Finally, being conducted in a single district of Delhi, the findings may not be representative of all urban poor populations across India. Also, the lower proportion of the geriatric population in the study area (6.9%) compared to the national average (~10%) limits the applicability of our findings to this key high-risk group for antibiotic misuse. Finally, a limitation of this study is that it did not assess participant exposure to antibiotic and AMR awareness campaigns, which precludes an evaluation of the effectiveness of such public health initiatives.

The present study has important public health implications. First, the evidence from this study clearly indicates that the conceptual comprehension of AMR and perceived individual risk is largely absent within the community, with concomitant high levels of self-medication. Further, there has likely been no perceptible improvement in antibiotic KAP in the past decade, especially in vulnerable low socioeconomic status populations. The undesirable behaviors are perpetuated by public belief in the power of antibiotics to address multiple health problems, including those of viral or non-infectious etiology. Second, the fact that only a third of participants understood the importance of completing the prescribed antibiotic course highlights a critical failure in patient education, which may stem from both patient-related factors especially low health literacy, and also ineffective communication from healthcare providers [28] Third, a crucial finding of this study is the extremely high prevalence of ‘Don’t know’ responses to questions about antibiotic knowledge pointing towards a comprehensive absence of information, not merely knowledge deficit. This is a significant insight, as addressing an information vacuum with targeted education is relatively more straightforward than correcting long-held misinformation. The demonstrated impact of mass media campaigns on improving antibiotic knowledge and attitudes in European countries supports the feasibility of achieving similar positive outcomes also in the Indian scenario [29,30,31]. Consequently, effective AMR-related health communication must be urgently deployed to curb antibiotic misuse and nudge behavior change, which requires using both mass media and small-group communication directly within communities and health facilities. To ensure equity, interventions must be gender-sensitive, adapted for low educational levels, and leverage existing community platforms frequented by women, such as Anganwadis.

In conclusion, critical deficiencies exist regarding knowledge about antibiotics and AMR, coupled with a high prevalence of inappropriate use, within a low-income urban community in Delhi. Poor antibiotic and AMR KAPs are deeply linked with adverse social determinants, particularly low educational levels, economic disparities, and gender inequalities, which frequently preclude the safe and effective use of antibiotics. Developing and deploying evidence-informed interventions is a critical necessity for curbing irrational antibiotic demand and thus preserving these vital medicines for future generations. Furthermore, such educational and behavioral change strategies must be meticulously tailored to community demographics, including gender, literacy, and socioeconomic status.

## 4. Methods

Design and Setting: A community-based cross-sectional study was conducted from April to June 2025 for a demographic, health and environmental surveillance site and the field practice area of a government medical college in the North-East district of Delhi. The area includes a mixed population from an urban resettlement colony, a slum, and a smaller urban village with a cumulative population of 54,614 as per the local census. Due to high levels of pre-existing community engagement, a high response rate to the survey was achieved (95%). The detailed sociodemographic characteristics of the population have been reported elsewhere [32].

Selection criteria: The participants were aged 18 years and above, and residents of the study site for at least six months, and had comprehension of the local languages, Hindi or English. We excluded individuals who were healthcare professionals or students in medical, pharmaceutical, or related fields (to focus on the general population’s awareness). Individuals with significant cognitive impairments or language barriers preventing effective communication were also excluded from the survey.

### 4.1. Study Outcomes

The Primary Outcome of this study was the proportion of participants demonstrating adequate knowledge and awareness of antimicrobial resistance (AMR) and appropriate antibiotic use practices.

Secondary Outcomes include sociodemographic factors (age, gender, education, socioeconomic status) associated with higher or lower levels of knowledge and awareness of AMR; and the prevalence of behaviors contributing to antibiotic misuse, including the proportion of participants engaging in self-medication with antibiotics, and the proportion of participants failing to complete prescribed antibiotic courses.

### 4.2. Sample Size and Sampling

The sample size estimation was based on the primary objective, which is to determine the prevalence of knowledge of AMR among the general population in Delhi. According to the study conducted by Bhardwaj et al. (2022), the prevalence of poor knowledge of antibiotics in the general population in India was 49% [17]. Therefore, the minimum sample size calculated using the Cochran formula (N = (Z^2^pq)/d^2^XDeff) at 99% confidence levels, 5% absolute precision, design effect of 2 (to correct the loss of variance), and anticipating 10% non-response was 1458. To ensure our study was sufficiently powered, we set our final target sample size at 1600 [33,34].

The sampling frame included all the residential households within the study site. The entire study area was divided into 16 sectors (clusters) based on the neighborhood geographical profile, of which four were selected by the simple random sampling method. Within each selected cluster, an equal number of households were selected using the systematic random sampling method. For multi-person households, a single respondent was randomly selected using the Kish Selection Grid method. This process was executed by a computer-assisted algorithm that used the total number of household adults and their sequential age ranking as selection criteria. If the selected respondent was unavailable or unwilling to participate, then a follow-up visit was undertaken at a mutually agreed-upon time for a total of two additional attempts. If unavailable after repeated attempts, the household was replaced by the next eligible household in the sampling frame.

Methodology: Data were collected by eight trained field investigators through face-to-face interviews with the participants in the local language, Hindi. The study adapted the validated survey instrument developed by WHO for their multi-country survey (2015) to assess participants’ KAP regarding antibiotic use and antimicrobial resistance [9]. Responses were collected on multiple-choice and Likert-scale items based on the standardized tool. The survey instrument was translated into Hindi following a standardized back-translation protocol to ensure linguistic and cultural accuracy. Subsequently, the instrument was supplemented with questions on sociodemographic characteristics. The participants were shown to have commonly dispensed antibiotic formulations at the beginning of the survey to facilitate identification and recall. Data were collected through face-to-face interviews by trained field investigators and recorded electronically using Android tablets and stored securely in a password-protected database. All data was anonymized to ensure confidentiality.

### 4.3. Measurement of Outcomes

(i).Knowledge Assessment: A total of 13 questions were used to compute the knowledge score. Each response was dichotomously coded, with correct answers assigned a value of 1 and other responses (including wrong True/False selections and “Don’t know”/”Can’t remember” answers) coded as 0. This approach yielded a total knowledge score ranging from 0 to 13 points, with higher scores indicating greater antibiotic-related knowledge (Cronbach’s alpha, α = 0.80).(ii).Attitude assessment: Participants’ attitude was assessed using 14 statements (presented in Figure 1 and Figure 2) measured on a 5-point Likert scale ranging from 1 (“Strongly disagree”) to 5 (“Strongly agree”). The total attitude score was calculated by summing responses across all statements, yielding a possible range of 14 to 70 points. Higher scores indicated more positive attitudes toward appropriate antibiotic use and antimicrobial resistance management (Cronbach’s alpha, α = 0.85).(iii).Practice assessment: A total of 5 practice-related questions were used to generate a practice score, which included behaviors such as obtaining prescriptions from a physician when last used antibiotics, receiving advice from a physician on antibiotic use, purchasing medicines from a medical store, self-medication, and checking medicine expiry. Each reported practice was scored dichotomously, with appropriate behaviors coded as 1 and inappropriate practices or cannot remember responses coded as 0, yielding a total score ranging from 0 to 5, where higher scores indicated more medically appropriate behaviors (Cronbach’s alpha, α = 0.72).

For each KAP domain, total scores were further categorized into ordinal levels using the following percentile-based thresholds: Poor: Minimum to ≤25th percentile (coded as 0); Moderate: >25th to ≤75th percentile (coded as 1); Good: >75th percentile (coded as 2).

Statistical analysis: Descriptive statistics were used to summarize sociodemographic characteristics and KAP scores. Categorical variables were reported as frequencies and percentages. Ordinal logistic regression was used to assess the association between sociodemographic variables and the categorized knowledge, attitude, and practices scores, respectively. Univariable ordinal logistic regression models were initially performed to identify variables significantly associated with the outcomes, and those variables were included in the adjusted models. The proportional odds assumption for the ordinal models was tested using the Brant test. In cases where this assumption was violated, generalized ordered logistic regression (gologit2) was performed. No multicollinearity was observed among the independent variables included in the model. All regression results are reported as adjusted odds ratios (aOR) with 95% confidence intervals (CI), and a *p*-value of <0.05 was considered statistically significant. Data analysis was performed using Stata version 15.1 (StataCorp., College Station, TX, USA).

### 4.4. Ethical Considerations

The study was approved by the Institutional Ethics Committee (F.1/IEC/MAMC/14/02/2025/No89) and prospectively registered with the Clinical Trial Registry of India as an observational study (CTRI/2025/05/087858). Informed consent was provided by all the participants. After the data collection process, multiple health awareness sessions focusing on antimicrobial resistance (AMR) and the responsible use of antibiotics were conducted in the community.

## Figures and Tables

**Figure 1 antibiotics-14-01184-f001:**
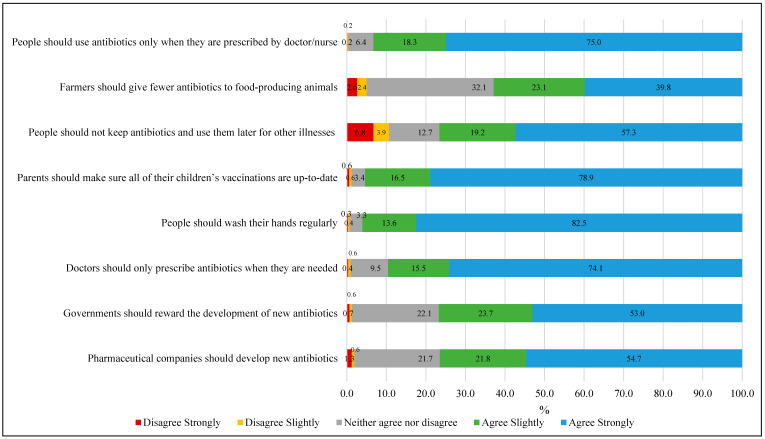
Attitude of participants towards antibiotic use and antimicrobial resistance (N = 1601).

**Figure 2 antibiotics-14-01184-f002:**
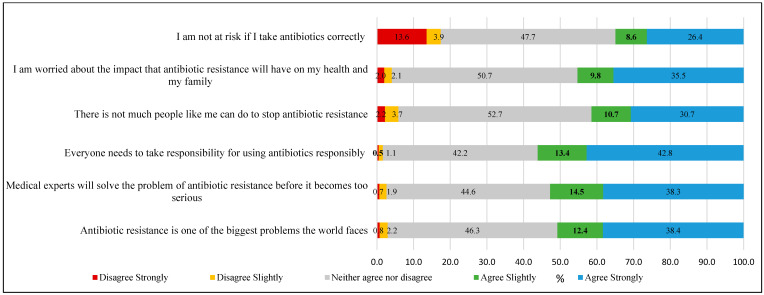
Perceptions on risk and responsibility regarding antimicrobial resistance (N = 1601).

**Table 1 antibiotics-14-01184-t001:** Sociodemographic characteristics of the participants.

	N = 1601
	*n* (%)
Age, mean (SD)	34.7 (13.7)
Gender	
Male	697 (43.5)
Female	904 (56.5)
Education	
Below high school	323 (20.2)
High school graduate	189 (11.8)
12th grade with technical/vocational training/associate degree	549 (34.3)
Bachelor’s degree	398 (24.9)
Master’s/Professional degree/Doctorate	142 (8.9)
Total household income (monthly)	
Below ₹10,000	188 (11.7)
₹10,001–₹25,000	793 (49.5)
₹25,001–₹50,000	367 (22.9)
Above ₹50,001	253 (15.8)
Household size, mean (SD)	5.4 (2.1)

**Table 2 antibiotics-14-01184-t002:** Knowledge of antibiotics and antimicrobial resistance among participants.

	N = 1601
	*n* (%)
Knowledge of antibiotics	
Heard/know of antibiotics	1088 (68.0)
When should antibiotics be stopped once started?	
When you’ve taken all of the antibiotics as directed	556 (34.7)
When you feel better	190 (11.8)
Don’t know	855 (53.4)
Okay to use antibiotics prescribed to a friend/family member if used to treat the same illness	
True	380 (23.7)
False	1044 (65.2)
Don’t know	177 (11.1)
Okay to buy the same antibiotics, or request from a doctor, for a similar illness if it has helped get better in the past	
True	388 (24.2)
False	989 (61.8)
Don’t know	224 (14.0)
Knowledge about antibiotic resistance	
Heard of the following terms *	
Antibiotic resistance	147 (9.2)
Superbugs	29 (1.8)
Antimicrobial resistance	42 (2.6)
AMR	67 (4.2)
Drug resistance	143 (8.9)
Antibiotic-resistant bacteria	119 (7.4)
None	1291 (80.6)
Antibiotic resistance occurs when your body becomes resistant to antibiotics, and they no longer work as well	
True	517 (32.3)
False	143 (8.9)
Don’t know	941 (58.8)
Many infections are becoming increasingly resistant to treatment with antibiotics	
True	502 (31.4)
False	104 (6.5)
Don’t know	995 (62.1)
If bacteria are resistant to antibiotics, it can be very difficult or impossible to treat the infections they cause	
True	540 (33.7)
False	133 (8.3)
Don’t know	928 (58.0)
Antibiotic resistance is an issue that could affect me or my family	
True	594 (37.1)
False	128 (8.0)
Don’t know	879 (54.9)
Antibiotic resistance is an issue in other countries, but not here	
True	260 (16.2)
False	363 (22.7)
Don’t know	978 (61.1)
Antibiotic resistance is only a problem for people who take antibiotics	
True	353 (22.1)
False	331 (20.7)
Don’t know	916 (57.2)
Bacteria that are resistant to antibiotics can be spread from person to person	
True	526 (32.9)
False	195 (12.2)
Don’t know	880 (54.9)
Antibiotic-resistant infections could make medical procedures like surgery, organ transplants and cancer treatment much more dangerous	
True	417 (26.1)
False	135 (8.4)
Don’t know	1049 (65.5)

* Indicates a multiple response.

**Table 3 antibiotics-14-01184-t003:** Antibiotic Use and Self-Medication practices among participants.

	N = 1601
	*n* (%)
Use of antibiotics	
The last time antibiotics were taken (N = 1601)	
In the last month	266 (16.6)
In the last 6 months	227 (14.2)
In the last year	90 (5.6)
More than a year ago	143 (8.9)
Never	236 (14.7)
Can’t remember	639 (39.9)
On that occasion, they had received antibiotics or a prescription from a doctor/nurse (N = 1365 ^a^)	651 (47.7)
On that occasion, they had received advice from a doctor/pharmacist on how to take them (N = 1365 ^a^)	663 (48.6)
On that occasion, antibiotics were obtained from (N = 1365 ^a^)	
Medical store	549 (40.2)
Other	38 (2.8)
Can’t remember	781 (57.2)
% that self-medicate with antibiotics (N = 1601)	394 (24.6)
Common symptoms for which participants self-medicate (N = 394) *	
Headache	153 (38.8)
Fever	313 (79.4)
Cough	157 (39.8)
Pain	182 (46.2)
Diarrhea	16 (4.1)
Acidity	68 (17.3)
Others	7 (1.8)
Sources of information about self-medication (N = 368)	
Internet	90 (24.5)
Friends/Family	145 (39.4)
Healthcare professionals	276 (75.0)
Old prescription	39 (10.6)
Advertisement	24 (6.5)
Neighbor	10 (2.72)
Check the expiry of the medication (N = 1601)	975 (60.9)

^a^ Individuals who responded “Never” to “The last time antibiotics were taken” were excluded from the denominator. * Indicates multiple responses.

**Table 4 antibiotics-14-01184-t004:** Sociodemographic Correlates of Knowledge and Practices of Antibiotic Use and Antimicrobial Resistance.

Variable	Crude OR (95% CI)	Adjusted OR ^a^ (95% CI)	*p*-Value	Adjusted OR ^b^ (95% CI)	*p*-Value
I. Knowledge		N = 1598		N = 1598	
Age	0.97 (0.96, 0.98)	0.99 (0.98, 1.00)	0.003	0.99 (0.98, 1.00)	0.003
Gender					
Male	Ref	Ref		Ref	
Female	0.65 (0.54, 0.79)	0.82 (0.67, 1.00)	0.05	0.82 (0.67, 1.00)	0.05
Education					
Below high school	0.09 (0.07, 0.13)	0.16 (0.11, 0.22)	<0.001	0.16 (0.11, 0.22)	<0.001
High school	0.40 (0.29, 0.56)	0.38 (0.26, 0.57)	<0.001	0.60 (0.40, 0.90)	0.012
12th grade	0.31 (0.24, 0.39)	0.35 (0.27, 0.46)	<0.001	0.35 (0.27, 0.46)	<0.001
College degree and above	Ref	Ref		Ref	
Income					
<₹10,000	0.14 (0.10, 0.20)	0.21 (0.12, 0.37)	<0.001	0.21 (0.12, 0.37)	0.001
₹10,001–₹25,000	0.38 (0.29, 0.50)	0.33 (0.20, 0.54)	<0.001	1.20 (0.84, 1.70)	0.317
₹25,001–₹50,000	0.70 (0.52, 0.95)	0.49 (0.29, 0.83)	0.008	1.41 (0.98, 2.04)	0.066
>₹50,000	Ref	Ref		Ref	
II. Attitude		N = 1598		N = 1598	
Age	0.98 (0.97, 0.99)	0.98 (0.97, 0.99)	<0.001	0.98 (0.97, 0.99)	<0.001
Gender					
Male	Ref				
Female	0.96 (0.79, 1.16)				
Education					
Below high school	0.56 (0.42, 0.74)	2.36 (1.63, 3.41)	<0.001	2.36 (1.63, 3.41)	<0.001
High school	0.88 (0.63, 1.22)	1.51 (1.04, 2.19)	0.03	1.51 (1.04, 2.19)	0.03
12th grade	0.33 (0.26, 0.42)	0.53 (0.40, 0.70)	<0.001	0.53 (0.40, 0.70)	<0.001
College degree and above	Ref	(1.00) Ref		(1.00) Ref	
Income (per month)					
<₹10,000	0.09 (0.06, 0.13)	0.10 (0.06, 0.17)	<0.001	0.10 (0.06, 0.17)	<0.001
₹10,001–₹25,000	0.58 (0.43, 0.77)	0.71 (0.51, 1.00)	0.047	0.71 (0.51, 1.00)	0.047
₹25,001–₹50,000	0.68 (0.49, 0.94)	0.77 (0.54, 1.10)	0.147	0.77 (0.54, 1.10)	0.147
>₹50,000	Ref	(1.00) Ref		(1.00) Ref	
Knowledge of antibiotics and antimicrobial resistance	1.21 (1.17, 1.26)	1.28 (1.21, 1.35)	<0.001	1.05 (1.00, 1.11)	0.075
III Practice		N = 1363		N = 1363	
Age	1.00 (0.99, 1.00)				
Gender					
Male	Ref				
Female	0.93 (0.76, 1.13)				
Education					
Below high school	0.39 (0.30, 0.52)	0.98 (0.67, 1.42)	0.902	0.57 (0.35, 0.93)	0.024
High school	0.41 (0.30, 0.58)	0.89 (0.59, 1.34)	0.579	0.28 (0.14, 0.58)	0.001
12th grade	0.78 (0.60, 1.00)	1.02 (0.76, 1.36)	0.91	1.02 (0.76, 1.36)	0.91
College degree and above	Ref	Ref			
Income					
<₹10,000	0.69 (0.47, 1.02)	0.58 (0.35, 0.97)	0.037	0.94 (0.56, 1.58)	0.814
₹10,001–₹25,000	0.28 (0.21, 0.39)	0.37 (0.26, 0.53)	<0.001	0.37 (0.26, 0.53)	<0.001
₹25,001–₹50,000	0.47 (0.33, 0.67)	0.52 (0.35, 0.75)	0.001	0.52 (0.35, 0.75)	0.001
>₹50,000	Ref	Ref			
Knowledge score	1.14 (1.11, 1.18)	1.30 (1.23, 1.37)	<0.001	1.08 (1.02, 1.15)	0.01
Attitude score	0.95 (0.94, 0.96)	0.93 (0.92, 0.95)	<0.001	0.93 (0.92, 0.95)	<0.001

Knowledge, attitude, and practices are classified into three categories: poor (coded as 0), moderate (1), and good (2), respectively; higher categories indicate better knowledge, attitudes, and practices. ^a^ Represents adjusted odds ratio for poor vs. (moderate and good); ^b^ Represents adjusted odds ratio for (poor and moderate) vs. good.

## Data Availability

The anonymized dataset for this study will be made available on reasonable request to the corresponding authors.

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
