# Peer review of "Knowledge, Attitudes, and Practices of Antibiotic Use and AMR in Low-Income Urban Delhi, India: A Community-Based Cross-Sectional Study"

_antibiotics, 2025, doi:10.3390/antibiotics14121184_

Round 1
Reviewer 1 Report
Comments and Suggestions for Authors
Dear authors
I enjoyed the opportunity to review your manuscript entitled ‘Knowledge , Attitudes, and Practices of Antibiotic Use and AMR in Low-Income Urban Delhi, India: a community-based cross-sectional study’. The manuscript was well constructed, easy to follow, and the topic is of importance and suitable for the readership of Antibiotics. It is imperative that we understand public knowledge and behaviours in order to inform future messaging on AMR. I particularly found the methods section very clearly drafted and the statistical methodology easy to follow, thank you.
I do however have a few issues to be address before the manuscript can be accepted for publication
Introduction
Line 38: I suggest adding the words ‘and have’ after medical costs
Line 43: I suggest using the latest reference for this work which is Naghavi, Mohsen et al. Global burden of bacterial antimicrobial resistance 1990–2021: a systematic analysis with forecasts to 2050. The Lancet, Volume 404, Issue 10459, 1199 – 1226. They state Our forecasts show that an estimated 1·91 million (1·56–2·26) deaths attributable to AMR and 8·22 million (6·85–9·65) deaths associated with AMR could occur globally in 2050.
Line 59: This reference relates to work carried out in the UK and does not talk about the issues in India as suggested by the sentence it is referencing.
Methods
Line 93: Please reference where the census data is hosted
Query: Were participants renumerated for participation?
Results
Line 195: In my experience, albeit UK based, it is unusual for the majority of respondents to be in the younger age bracket although this maybe due to study design. My query however is how do you think this might have affected the outcome as the younger age group have less lived experience? It would be interesting to carry out regression analysis on schooling, age and knowledge.
Line 204: The statement 24.2% would repurchase antibiotics without a medical consultation appears misleading as the actual question in table 2 states “Okay to buy the same antibiotics, or request from a doctor, ….” Please rectify.
Table 2: The majority of respondents have selected Don’t know for all knowledge questions. This should be emphasised more in the discussion as it suggests that more education could improve this knowledge.
Discussion
Line 227 – 280: I’m not sure I see how your cohort having lower AMR knowledge definitely “indicates a high prevalence of inadequate function health literacy…” You may want to say that In line with findings from Muflih et al., this may suggest….
Line 280 – 282: The cohort is described as coming from a socioeconomically vulnerable population yet this statement discusses the general public. If the authors want to compare their findings to the general population then they must clarify how and why you feel that your cohort are relatable to the general population e.g.is there national data on the percentage of the population who go to school, what the national reading age is, national socio-demographic profile, etc.
A second point here - Is it possible that they may not have access to conventional public health messaging on AMR? This should be clarified.
Author Response
Reviewer 1
Comment 1 Line 38: I suggest adding the words ‘and have’ after medical costs
Response 1. Complied.
Comment 2 Line 43: I suggest using the latest reference for this work which is Naghavi, Mohsen et al. Global burden of bacterial antimicrobial resistance 1990–2021: a systematic analysis with forecasts to 2050. The Lancet, Volume 404, Issue 10459, 1199 – 1226. They state Our forecasts show that an estimated 1·91 million (1·56–2·26) deaths attributable to AMR and 8·22 million (6·85–9·65) deaths associated with AMR could occur globally in 2050.
Response 2. Complied. We have updated the reference
Comment 3 Line 59: This reference relates to work carried out in the UK and does not talk about the issues in India as suggested by the sentence it is referencing.
Response 3. We have now replaced reference 10 with an Indian reference. Reference to AMR as a silent pandemic is now mainstream and applicable to most regions worldwide.
Methods
Comment 4. Line 93: Please reference where the census data is hosted
Response 4. This local census data is not available in the public domain but is maintained by the Department of Community Medicine, Maulana Azad Medical College, New Delhi and utilized for public health research and practice. However, the aggregate population characteristics have been reported previously as part of the six demographic environmental surveillance sites developed in medical colleges and research institutions from across the country (Reference 19).
Comment 5. Query: Were participants renumerated for participation?
Response 5. We understand you imply whether the participants were provided any monetary compensation for their time. No, the participants were not provided with any financial compensation to participate in this study. This is in accordance with our institutional ethics committee's policy, which prohibits such practices to avoid the potential for undue inducement, particularly for participants from lower socioeconomic backgrounds
Results
Comment 6. Line 195: In my experience, albeit UK based, it is unusual for the majority of respondents to be in the younger age bracket although this maybe due to study design. My query however is how do you think this might have affected the outcome as the younger age group have less lived experience? It would be interesting to carry out regression analysis on schooling, age and knowledge.
Response 6. We agree that the study population predominantly constitutes that from the working age-group. In our study, we had 124 (7.7%) geriatric (age≥60) participants, sampled nearly proportionate to the geriatric population in our study area (6.9%). This is consistent with evidence that slums and resettlement colonies in the cities, often comprise significant youthful migrant populations from rural regions and suburbs looking for economic and educational opportunities. We have also acknowledged the following limitation “The lower proportion of the geriatric population in the study area (6.9%) compared to the national average (~10%) limits the applicability of our findings to this key high-risk group for antibiotic misuse.”
We have already included age and schooling in the regression model (Table 4) (Line 231-250). Those with higher educational attainments (graduate and above) had significantly higher odds of having good knowledge compared to those with comparatively lower educational attainments. Furthermore, those with upto high school education, had significantly lower odds of engaging in better practices compared to those with high educational attainments (college degree or above). We have also revised the terminology used to describe educational attainment to ensure consistency with international standards.
Comment 7. Line 204: The statement 24.2% would repurchase antibiotics without a medical consultation appears misleading as the actual question in table 2 states “Okay to buy the same antibiotics, or request from a doctor, ….” Please rectify.
Response 7. Complied. The line has been rephrased as per the actual question to prevent any misinterpretation “24.2% indicated they would repurchase antibiotics or request them from a doctor for a similar illness, provided the treatment was perceived to be effective previously”
Comment 8. Table 2: The majority of respondents have selected Don’t know for all knowledge questions. This should be emphasised more in the discussion as it suggests that more education could improve this knowledge.
Response 8. We thank the reviewer for this excellent suggestion. We agree that the high prevalence of 'Don't know' responses is a key finding that warrants greater emphasis in the discussion. We have revised the section to not only highlight this but also to discuss its specific implications—namely, that this indicates an absence of information rather than misinformation, which strongly supports the potential for successful educational interventions.
The revised text in the Discussion section now reads as follows
“Third, a crucial finding of this study is the extremely high prevalence of 'Don't know' responses to questions about antibiotic knowledge pointing towards a comprehensive absence of information, not merely knowledge deficit. This is a significant insight, as ad-dressing an information vacuum with targeted education is relatively more straightforward than correcting long-held misinformation. The demonstrated impact of mass media campaigns on improving antibiotic knowledge and attitudes in European countries supports the feasibility of achieving similar positive outcomes also in the Indian scenario”
Discussion
Comment 9. Line 227 – 280: I’m not sure I see how your cohort having lower AMR knowledge definitely “indicates a high prevalence of inadequate function health literacy…” You may want to say that In line with findings from Muflih et al., this may suggest….
Response 9. Complied. We have rephrased as “In line with the findings of Muflih et al (2021), this may suggest that many individuals lack the functional health literacy required to obtain, process, and comprehend information regarding the rational use of antibiotics”
Comment 10. Line 280 – 282: The cohort is described as coming from a socioeconomically vulnerable population yet this statement discusses the general public. If the authors want to compare their findings to the general population then they must clarify how and why you feel that your cohort are relatable to the general population e.g.is there national data on the percentage of the population who go to school, what the national reading age is, national socio-demographic profile, etc.
Response 10. We thank the reviewer for this insightful comment and for highlighting the need to better contextualize our findings. The reviewer is correct that a comparison between our specific cohort and the general population must be carefully justified.
To provide the most current and relevant data to address the reviewer's point, we have updated our comparison using the latest Periodic Labour Force Survey (PLFS):
Consequently, we have revised the statement in the manuscript in the following manner
“This study affirms the persistence of a significant antibiotic knowledge deficit within a dense urban-slum and resettlement colony population. Notably, while 79.8% of re-spondents had completed at least a high-school education, their knowledge gaps re-mained. The educational attainment of our sample is considerably higher than India's current adult literacy rate (80.9%, per the 2023-24 Periodic Labour Force Survey) [24]. We attribute this discrepancy to the lower proportion of geriatric participants in our study compared to the national average, as this demographic typically has less formal education and overall lower educational attainments.”
Comment 11. A second point here - Is it possible that they may not have access to conventional public health messaging on AMR? This should be clarified.
Response 11. We thank the reviewer for raising this crucial point. The suggestion that our study population may have limited or differential access to conventional public health messaging on AMR is highly pertinent and could indeed be a significant contributing factor to our findings. Consequently, in the discussion section, we have clarified that “The knowledge gaps observed in the participants may also be compounded by barriers to accessing conventional public health messaging and failure to tailor public health campaigns resulting in potential lack of effective outreach”
We have also added the following point acknowledging lack of this data in the limitations section “Finally, a limitation of this study is that it did not assess participant exposure to antibiotic and AMR awareness campaigns, which precludes an evaluation of the effectiveness of such public health initiatives.”
Reviewer 2 Report
Comments and Suggestions for Authors
The manuscript “Knowledge, Attitudes, and Practices of Antibiotic Use and AMR in Low-Income Urban Delhi, India: a community-based cross-sectional study” by Rao et al. performs a cross-sectional study to look for awareness deficits and high-risk antibiotic use practices in low-income communities in Delhi, India.
The manuscript presents very interesting and complete data about people’s knowledge and practices concerning antibiotic use and antibiotic resistance. The text, as well as the methods and data presented, are very clear.
I have a few minor comments.
- I think there is a problem with the use of Cochran formula. Assuming p = 49% = 0.49; q = 1-p = 0.51, Z = 1.96 (because 95% CI), d = 2.5% = 0.025, and Deff = 2, gives N = 3072, not 1520. Perhaps I am wrong somewhere or, in fact, perhaps there is no need to consider Deff=2. Or perhaps d = 5% and Deff = 4.
- There is a problem with the sentence in lines 50-51 “A WHO multi-country…(AMR)”
- Line 179: odds ratio (OR)? adjusted odds ratio (AOR)?
- Line 247: missing “, ” in “CI: 0.920.95” instead of “CI: 0.92, 0.95”
- Line 266: what is LMICs, Low- and Middle-Income Countries?
- Line 266: Gualano, not Guanalo
- Lines 266 – 268. In the study by Gualano [ref. 7], please specify who (e.g., which countries) was unaware that antibiotics treat bacterial infections, etc.
Author Response
Comment 1. I think there is a problem with the use of Cochran formula. Assuming p = 49% = 0.49; q = 1-p = 0.51, Z = 1.96 (because 95% CI), d = 2.5% = 0.025, and Deff = 2, gives N = 3072, not 1520. Perhaps I am wrong somewhere or, in fact, perhaps there is no need to consider Deff=2. Or perhaps d = 5% and Deff = 4.
Response 1. We are grateful to the reviewer for their diligence in checking our sample size calculation and for prompting us to clarify our methodology. The reviewer is correct that using their assumed parameters would yield a different result. We calculated the sample size using a different set of assumptions, which we believe are well-justified for our research context.
Considering 99% confidence levels (Z=2.576) to ensure a higher degree of certainty, p=0.49, d=0.5 (absolute precision of 5%), Deff=2 (to correct for the loss of variance), and anticipating 10% non-response, the minimum sample size was calculated to be 1458. To ensure our study was sufficiently powered, we set our final target sample size at 1600. We have updated the manuscript to include this detailed breakdown for clarity.
Comment 2. There is a problem with the sentence in lines 50-51 “A WHO multi-country…(AMR)”
Response 2. Corrected the mistake “A 2015 WHO multi-country survey revealed significant misconceptions regarding antimicrobial resistance (AMR), with 76% of respondents incorrectly believing that resistance develops in the human body rather than in bacteria.”
Comment 3. Line 179: odds ratio (OR)? adjusted odds ratio (AOR)?
Response 3. Adjusted odds ratio, corrected. Thankyou
Comment 4. Line 247: missing “, ” in “CI: 0.920.95” instead of “CI: 0.92, 0.95”
Response 4. Corrected
Comment 5. Line 266: what is LMICs, Low- and Middle-Income Countries?
Response 5. Yes, we have now replaced the abbreviation with the full expression low-and-middle-income countries
Comment 6. Line 266: Gualano, not Guanalo
Response 6. Corrected.
Comment 7. Lines 266 – 268. In the study by Gualano [ref. 7], please specify who (e.g., which countries) was unaware that antibiotics treat bacterial infections, etc.
Response 7. Complied. We have added the following information to the manuscript “Furthermore, a majority of participants in studies from Jordan, New Zealand, the midwestern USA, and South Korea were unaware that antibiotics treat bacterial infections, while majorities in South Korea, Jordan, Malaysia, and New York City's Latino community incorrectly believed antibiotics treat the cold and flu.”